# Nurses’ Perceptions on the Role of Advanced Psychiatric Nurses in Mental Healthcare: An Integrative Review

**DOI:** 10.3390/ijerph22040626

**Published:** 2025-04-16

**Authors:** Femi Edward Duyilemi, Nkhensani Florence Mabunda

**Affiliations:** 1Behavioral Health Interdisciplinary Program, Mental Care Line, Michael DeBakey VA Medical Center, Houston, TX 77030, USA; femi.duyilemi@va.gov; 2Department of Nursing Science, School of Health Care Sciences, Sefako Makgato Health Science University, Molotlegi St., Ga-Rankuwa Zone 1, Ga-Rankuwa 0208, South Africa

**Keywords:** advanced psychiatric nurses, clinical skills, specialized knowledge, patient care outcomes, registered nurses, role definition

## Abstract

**Background/Objectives:** Advanced psychiatric nurses’ clinical skills and expertise are increasingly challenging registered nurses in mental health. Understanding registered nurses’ perceptions of the role of advanced psychiatric nurses is vital for improving mental healthcare delivery and fostering collaboration for effective patient outcomes. This integrative review aims to explore how registered nurses perceive the role of advanced psychiatric nurses in mental healthcare. **Methods**: An integrative review methodology was used to synthesize the existing literature following PRISMA guidelines. Both qualitative and quantitative studies provided a comprehensive understanding of the registered nurses’ perceptions regarding the role of advanced psychiatric nurses in mental health nursing. **Results**: Several key themes emerged from studies included in this review, including recognition of expertise, role ambiguity and boundaries, and the need for structured education and training. **Conclusions**: This review highlights the need to clarify roles and how communication is essential for improving collaboration, team cohesion, and patient outcomes while promoting interprofessional education to optimize mental healthcare, in addition to bridging the knowledge gap or the discrepancy between registered nurses and advanced psychiatric nurses. What registered nurses know and what they need to know to perform tasks will improve the quality of mental healthcare and optimize services for individuals with mental health needs.

## 1. Introduction

The demand for mental health services, exacerbated by the increasing ongoing shortage of mental health professionals, has led to the expanded role of advanced psychiatric nurses (APNs) in the healthcare context globally. APNs are highly trained nurses providing specialized, high-quality psychiatric care services, from assessment and diagnosis to treatment planning and patient management, while registered nurses (RNs) are integral to the multidisciplinary care team [1]. However, despite the critical role of APNs, there remains a limited understanding of how RNs perceive the role and responsibilities of APNs in mental healthcare settings. This knowledge gap can influence collaboration, communication, and ultimately the quality of care delivered to individuals diagnosed with mental illness [2,3].

Mental health nursing plays a central role in addressing the mental health needs of individuals diagnosed with mental illness, with the remarkable rising global dominance of mental health disorders. APNs, with their expert knowledge and expanded clinical skills, have become mandatory members of mental healthcare teams. These experts are tasked with providing direct patient care and aiding in leadership and educational roles within the healthcare context [4,5]. Despite their increasing significance, the incorporation of these experts into multidisciplinary teams is a complex process, affected by various factors such as role uncertainty, interaction barriers, and interprofessional collaboration challenges [6].

Even though RNs are frontier providers of healthcare working closely with APNs, their perceptions of APNs in mental health settings are essential for improving the efficacy of psychiatric care teams. However, research on RNs’ perceptions of APNs’ roles is scarce, and it is fundamental to pay attention to how these perceptions can affect the reliability of interprofessional practices and role clarification in mental health nursing [7]. APNs, with their expertise in clinical training and specialized psychiatric knowledge, have been progressively relied upon to fill this gap, playing a crucial role in delivering comprehensive quality mental healthcare [8].

However, despite their expanding responsibilities, little is known about how RNs perceive APNs within the mental healthcare team and how they work together in clinical practice [9]. RNs, as essential multidisciplinary team members, often work alongside APNs, and their perceptions of APNs can significantly impact teamwork, communication, and patient outcomes in a mental health context [10,11].

The gap in research on this topic presents a fundamental challenge. Subsequently, recognizing RNs’ perceptions of APNs’ roles, expertise, and the scope of their responsibilities is significant for fostering a more mutual, effective, and unified approach to mental healthcare service delivery [12]. This integrative review aims to fill this gap to extend the existing literature to enhance collaboration, improve patient outcomes, and optimize the incorporation of APNs into psychiatric care teams [13].

## 2. Materials and Methods

An integrative review methodology was employed to explore the existing literature on RNs’ perceptions regarding the role of APNs in psychiatric care. This approach was selected because it allows for the inclusion of both qualitative and quantitative studies. Five steps, namely, (1) problem identification, (2) literature search, (3) data evaluation, (4) data analysis, and (5) presentation of findings, were used to provide a comprehensive understanding of the topic [14,15]. This method was considered appropriate because it is aligned with evidence-based practice for conducting integrative reviews, ensuring a rigorous and systematic approach [16].

### 2.1. Identification of the Research Problem

Against a background of increasing demand for mental health services, it is clear that there is an inadequate understanding of how RNs perceive the role of APNs’ scope of responsibilities [2], which may influence collaboration, communication, and patient outcomes in mental healthcare settings [3]. The research question of this review is “what are RNs’ perceptions of the role of APNs in mental healthcare”? However, there is a gap in the research presented by the lack of studies exploring how RNs view the role of advanced psychiatric nurses, their competencies, and how RNs interact with APNs in practice [11]. Understanding these perceptions is crucial for improving teamwork, enhancing the integration of advanced psychiatric nurses into healthcare teams, and optimizing the delivery of mental health services [12]. Consequently, there is a need to conduct an integrative literature review to examine the existing perception of RNs regarding the role of APNs and their responsibilities. In this review, variables and populations were identified from articles obtained from the initial search as described in the following subsections.

### 2.2. Literature Search

During the selection process, a search was used to identify articles using specific databases, such as PubMed, CINAHL, and PsycINFO, in September 2024. The search utilized a combination of keywords and Boolean operators; ‘AND’ and ‘OR’ were used to combine all concepts to capture relevant articles. The keywords search included “advanced psychiatric nurses”, “clinical skills”, “specialized knowledge”, “patient care outcomes”, “registered nurses”, and “role definition”. These keywords were used in various combinations to ensure the inclusion of studies that explored RNs’ perspectives on the roles and responsibilities of APNs within mental healthcare settings [4]. The search was limited to studies published in English-language articles [17] and engaged in all-inclusive readings of relevant full texts appropriate to the search question.

### 2.3. Inclusion and Exclusion Criteria

The inclusion and exclusion criteria for this integrative review were used to determine whether articles retrieved from the literature could be included in this study. The studies that focused on RNs’ perceptions of the roles and responsibilities of APNs were included. Additionally, the following were included: studies published between 2015 and October 2024, articles in English, and empirical studies (qualitative, quantitative, or mixed methods) with a clear focus on mental healthcare [18] directly relevant to the current healthcare environment.

The following were excluded: studies that primarily focused on the perspectives of APNs rather than RNs, including articles in the press, and conference proceedings as well as articles that did not specifically address mental healthcare or psychiatric nursing [19]. Additionally, included articles were selected by reading the titles and abstracts of all the relevant studies and excluding those that did not meet the inclusion criteria. Moreover, the PRISMA flow diagram from Page et al. [20] was then used to expand the screening of the studies for their relevance and potential inclusion, as indicated in Figure 1. The above steps were carried out under the supervision of experienced researchers to ensure rigor for this integrative review.

Figure 1 shows a PRISMA flow diagram summarizing the key stages of the study selection and inclusion/exclusion criteria, which typically include four stages (“Identification”, “Screening”, “Eligibility”, and “Included”), the process for screening studies based on Page et al. [20]. PubMed, 893; CINAHL, 757; PsyInfor, 783. Total: 2433 − 1206 = 122. 

### 2.4. Assessment of the Quality of the Articles

To make the quality evaluation clear and transparent, the two researchers summarized their findings. Each article selected was critically assessed for quality to ensure the overall integrity of this review. This involved evaluating each article against standardized criteria through a structured tool, the Critical Appraisal Skills Programme (CASP) checklist [21].

Table 1 shows the CASP checklist, a practical tool that consists of a series of questions that guide researchers in evaluating the rigor and applicability of research studies, helping them to judge the quality and relevance of research studies [21]. Both researchers rated each article on a numeric scale (i.e., 1–5) and quality level (i.e., “high”, “medium”, or “low”), depending on the criteria as indicated in Table 1.

In this study, an integrative review methodology is used to promote a holistic understanding of the research and synthesize findings from a variety of sources within healthcare [29]. It often identifies gaps in knowledge and provides direction for future research, particularly in fields like nursing, where diverse methodologies are used to study complex phenomena. Unlike systematic reviews, which focus on a narrower set of studies, an integrative review allows for the inclusion of a wider range of study designs. Integrative reviews can incorporate findings from various research designs, including experimental, non-experimental, qualitative, and case studies focusing on a more restricted set of study types [30].

## 3. Results

A data extraction process was used to gather comprehensive, appropriate, and precise data from all the included studies to compare, synthesize, and interpret the findings. This rigorous process was conducted to ensure the validity and reliability of this review’s conclusions [22].

Table 2 shows the data extracted systematically from the selected studies using a standardized data extraction form. Key information including the authors, aims, designs, settings, population, and summaries of the findings are recorded [31] as indicated in Table 2. This allowed for an efficient comparison of the studies and ensured that relevant information was captured for analysis [32,33].

From the eight studies reviewed, several key themes emerged regarding RNs’ perspectives on the role of APNs in mental healthcare across the studies analyzed (published between 2015 and 2024). Recurring themes emerged from the analysis of RNs’ perspectives on APNs in healthcare settings. These themes reflect the aspects associated with interprofessional relationships and the roles of RNs and APNs within multidisciplinary teams, which include recognition of expertise, role ambiguity and boundaries, and the need for structured education and training.

*Recognition of Expertise*: RNs consistently acknowledged the clinical competence, leadership abilities, and specialized knowledge of APNs. Subsequently, RNs appreciated the expert-level knowledge and advanced skills that APNs offer to the healthcare team, particularly in the management of complex psychiatric cases. RNs considered diagnostic assessment, psychopharmacology, and psychotherapy specialized skills of APNs as essential to improving patient outcomes [34]. Research shows that RNs often remark on the value APNs add to comprehensive treatment plans, particularly in managing cases that require a versatile approach and clinical expertise with a deep understanding of patient needs and outcomes [35]. Additionally, RNs also recognized APNs’ leadership roles, such as leading clinical initiatives, mentoring all nursing staff, and helping them to advocate for patients, ultimately enhancing clinical outcomes [36] and thus promoting a culture of continuous advance in patient care.

*Role Ambiguity and Boundaries*: Despite acknowledging the contributions of APNs, RNs frequently expressed concerns about role ambiguity. There was often confusion about the division of responsibilities and unclear boundaries between RNs’ roles and those of APNs. This uncertainty has led to tensions and difficulties when collaborating within multidisciplinary teams. Despite the clear recognition of APNs’ contributions, the theme of role ambiguity and boundaries between the roles of RNs and APNs was identified. Research shows that RNs feel that the division of responsibilities between the two roles and the scope of practice of APNs often lead to confusion in the workplace, particularly when duties and responsibilities overlap between the two professional groups [37]. Additionally, such ambiguity creates conflict within multidisciplinary teams, as these uncertain boundaries could lead to challenges in collaboration and incompetent care delivery [38].

*Need for Structured Education and Training*: RNs highlighted the importance of structured education and training to better understand the roles of APNs. They believed that enhanced education could promote smoother teamwork and improve interprofessional collaboration. Research shows that RNs are impressed with the role and scope of APNs and the advanced skills required, such as those for assessment, treatment plans, and leadership in clinical practice, and feel more assured in their ability to work efficiently [39]. Although it is noted in the literature that structured training programs could help standardize the roles and accountabilities of both RNs and APNs, thus reducing role misperception and promoting improved coordination in care delivery [40], these educational programs could also enhance communication skills and foster reciprocal respect, which are needed to improve collaboration in healthcare teams [41].

Yet, the key findings revealed that RNs consistently acknowledge the clinical competence, leadership abilities, and specialized knowledge of APNs, which is crucial for fostering a collaborative environment, leading to improved patient outcomes. Hence, role ambiguity in mental healthcare can impact communication and the well-being of healthcare workers. Therefore, to address these challenges, there is a clear need for structured education and training programs [42]. Moreover, these themes underscore the deep dynamics between RNs and APNs and the essential need for accuracy, training, and mutual respect to improve team functionality and patient outcomes [43].

A thematic analysis was conducted to identify recurring themes and patterns across the studies [44]. Each study was reviewed in detail, and themes were extracted based on the findings related to RNs’ perspectives on the role of APNs in psychiatric care [45]. The analysis focused on identifying commonalities and differences in how RNs perceive the contributions of APNs, the challenges in collaboration, and the facilitators of effective teamwork in psychiatric settings [46]. Each study included in this review was assessed for quality using the Critical Appraisal Skills Programme (CASP) checklist [47] tool for both qualitative and quantitative studies [48,49,50]. This ensured that the findings were based on rigorous research and minimized the risk of bias [51].

Additionally, categories are created by systematically identifying recurrent patterns and concepts within the data from multiple studies, grouping similar codes to form main themes that represent key aspects of the research topic [52]. Codes were organized into meaningful clusters based on shared characteristics across the reviewed literature. In this study, this process often involved initial coding, then progressively grouping codes into categories, and finally refining those categories into overarching themes.

## 4. Discussion

The findings of this review provide valuable insights into how RNs perceive the role of APNs in psychiatric care. Overall, RNs recognize the significant contribution of APNs in enhancing care quality through their expertise, clinical skills, and leadership. However, challenges remain in integrating APNs fully into interdisciplinary teams due to role confusion and lack of clear communication. This issue fully supports the previous research indicating that role ambiguity between RNs and APNs is a barrier to collaboration in healthcare settings [53].

Building on this, the results of this integrative review highlight several important aspects regarding RNs’ perspectives on the role of APNs in psychiatric care. A common theme across the studies was the recognition of APNs’ clinical competence and expertise. RNs consistently acknowledged the value that APNs bring to managing complex psychiatric cases, enhancing patient care, and improving team effectiveness [54]. Indeed, APNs’ advanced skills are crucial for navigating the complexities of psychiatric care, especially as mental health needs continue to rise globally. This finding aligns with previous research that emphasizes the positive impact of APNs on patient outcomes through their leadership and specialized knowledge [28].

Yet, regardless of the acknowledged strengths, RNs also expressed concerns about role ambiguity, which was a significant barrier to effective collaboration. Studies have identified that unclear role boundaries between RNs and APNs created tensions and hindered teamwork [55]. This uncertainty can lead to miscommunication, task overlap, and reduced efficiency within multidisciplinary teams. These findings are consistent with the work of Lewis et al. [56], who found that role ambiguity between nursing professionals can cause friction and reduce overall team cohesion.

In light of these challenges, RNs emphasized the need for structured education and training to clarify the roles of APNs and improve interprofessional collaboration. This aligns with the recommendations from other studies, which argue that better role clarification and interprofessional education can foster smoother teamwork and enhance patient care outcomes [57]. Providing ongoing education about the roles of different professionals can mitigate role confusion and improve the overall functioning of psychiatric care teams.

Therefore, to bridge this gap, it is crucial to establish clear role definitions and foster interprofessional education. Educating RNs and APNs about each other’s roles and promoting collaborative practice could lead to more effective care delivery and improved outcomes for patients with psychiatric disorders [58]. It is alluded that while RNs recognized the contributions of APNs to psychiatric care, addressing role ambiguity and promoting structured education are essential to improving collaboration and patient outcomes. Further research should explore the long-term impact of integrating APNs into psychiatric teams, particularly regarding team dynamics and patient outcomes [37,38,49].

Subsequently, while RNs recognized the significant contributions of APNs to mental healthcare, there is a clear need to address role ambiguity and promote interprofessional education to strengthen collaboration and improve patient outcomes. In addition, further research should examine the impact of APN integration on patient care and team dynamics, even as the findings from this integrative review shed insight into the implications for scientific practice, education, policy, and advocacy in the sphere of APNs and mental healthcare. It is noted in the literature that the perceptions of nurses regarding APNs can inform how mental health services could be structured and how nurses could be equipped for the expanding roles in healthcare delivery [59]. Thus, health organizations should ensure that mental healthcare delivery is improving, especially in settings with limited access to mental health specialists working at their full capacity to help prevent hospital readmissions [60].

This review points out the reputation of specialized education and training for nurses aspiring to become mental health specialists. RNs acknowledged the need for educational programs that offer broad content on psychiatric nursing, sophisticated therapeutic interventions, and treatment plans for mental health disorders [61]. This would prepare nurses to work collaboratively with APNs and other healthcare professionals to ensure high-quality patient care outcomes [62]. In addition, the findings from this integrative review highlighted APNs’ roles as central to the advocacy for policies to sustain the inclusion of APNs in the mental health labor force. The review suggests that supportive policies and frameworks are necessary to ensure that APNs can fully utilize their expertise in clinical settings [63].

As a result, the following recommendations were proposed based on the findings from this integrative review to enhance the effectiveness, recognition, and integration of APNs in mental healthcare settings. RNs’ perceptions highlight the need for healthcare institutions to foster environments that employ fully utilize the skills and expertise of APNs. Thus, the heads of healthcare institutions should advocate for the integration of APNs into multidisciplinary teams.

Nevertheless, it is recommended that nursing educational institutions should expand their psychiatric nursing programs and curricula to provide more in-depth training, considering the significance of specialized skills in psychiatric nursing, to prepare nurses for the growing demands of mental healthcare. Continuous professional development and post-graduate education opportunities should be offered to ensure that RNs and APNs remain up to date with the latest advancements, to prepare them for leadership roles within the mental healthcare system.

## 5. Conclusions

This integrative review underscores the importance of understanding RNs perspectives on APNs’ roles in psychiatric care. While RNs recognize the expertise and leadership of APNs in managing complex cases, they face challenges related to role ambiguity and unclear boundaries, which hinder effective collaboration. This review highlights the need to clarify roles, improve communication, and promote interprofessional education to optimize mental healthcare. RNs also call for structured education to foster better understanding and teamwork between RNs and APNs.

Addressing these issues through role clarification and education is essential for improving collaboration, team cohesion, and patient outcomes. Future research should explore the long-term impact of APN integration on team dynamics, patient care outcomes, and workforce effectiveness. Bridging the gap between RNs and APNs will improve the quality of psychiatric care and ensure better services for individuals with mental health needs. Additionally, future studies should focus on examining the challenges faced by APNs in their role, including burnout, workload, and interprofessional conflicts, as well as examining the specific impact of APNs on patient outcomes in various mental health settings.

## 6. Limitations

Nevertheless, despite the fact that this integrative review offers constructive insights into RNs’ perspectives on the role of APNs in mental healthcare, numerous limitations ought to be acknowledged. The studies included in this review mostly represent the perspectives of RNs towards APNs in a mental healthcare setting, which may limit the generalizability of the results to other healthcare settings with different healthcare contexts. Another limitation is that the studies emphasize the need for structured education without detailing the specific educational interventions necessary to address role ambiguity and uphold an effective alliance between RNs and APNs.

## Figures and Tables

**Figure 1 ijerph-22-00626-f001:**
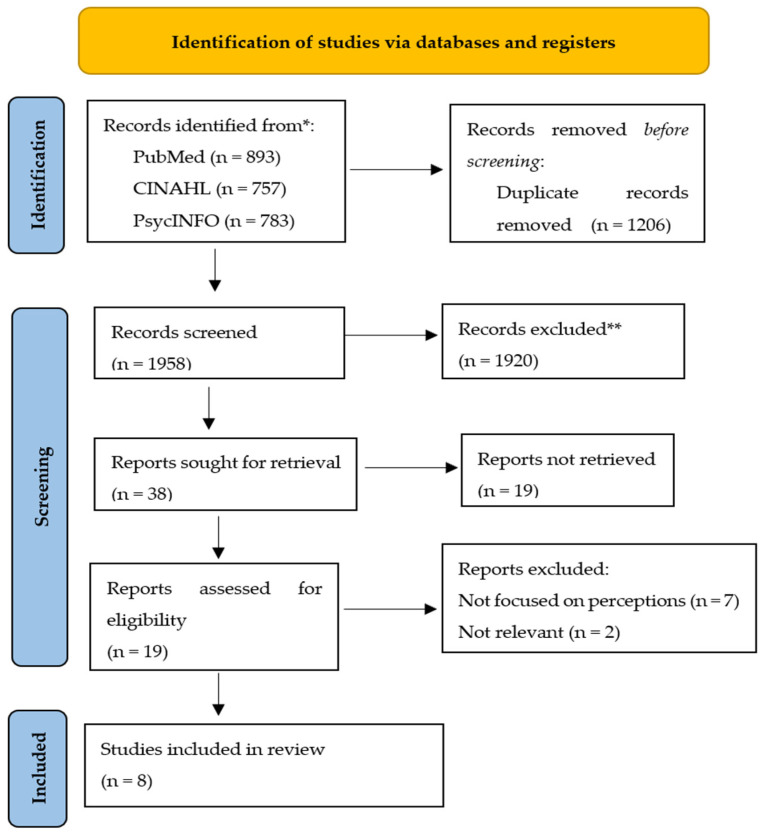
PRISMA flow diagram. Adapted from Page et al. The PRISMA 2020 statement: an updated guideline for reporting systematic reviews. *BMJ*. 29 March 2021; 372: n71. doi: 10.1136/bmj.n71. PMID: 33782057; PMCID: PMC8005924 [20]. In this study asterisks (* and **) were used to clarify the number of records screened and the reasons for exclusion.

**Table 1 ijerph-22-00626-t001:** Quality assessment of the articles.

Author and Year	Title	Relevance to Research Question	Data Analysis Quality (1–5)	Study Design Quality (1–5)	Overall Quality Rating	1st Rating	2nd Rating	Final Rating
[21]	The Distribution of Advanced Practice Nurses Within the Psychiatric Workforce	Low	3	2	Low	Low	Low	Low
[22]	Health Nurses in Integrated Care: Policy	Low	3	2	Low	Low	Low	Low
[23]	Facilitators of and Barriers to the Therapeutic Nurse-Patient Relationship: Perceptions From Psychiatric Mental Health Nurses	Medium	4	3	Medium	Medium	Medium	Medium
[24]	The Effective Use of Psychiatric Mental	High	5	4	High	High	High	High
[25]	Resilience and mental health nursing: An integrative review of international literature	Medium	4	3	Medium	Medium	Medium	Medium
[26]	Utilizing the mental health nursing workforce: A scoping review of mental health nursing clinical roles and identities	High	5	4	High	High	High	High
[27]	A global perspective of advanced practice nursing research: A review of systematic reviews	High	4	5	High	High	High	High
[28]	Effect of state regulatory environments on advanced psychiatric nursing practice	High	4	5	High	High	High	High

**Table 2 ijerph-22-00626-t002:** Summary of data extraction.

Author and Year	Title	Aim	Design	Setting	Population	Summary of Findings
Beck et al., 2020 [21]	The Distribution of Advanced Practice Nurses Within the Psychiatric Workforce	To examine the size and distribution of the advanced practice psychiatric nurse workforce relative to the total psychiatry workforce to determine whether nurses are predominantly working in areas with higher or lower numbers of behavioral health specialists.	Cross-sectional study	American Nurses Credentialing Center	Mental health psychiatric nurses, adult psychiatric nurses, child psychiatric clinical nurse specialists, and adult psychiatric clinical nurse specialists	The study found inconsistent patterns of how psychiatric nurses are distributed relative to the rest of the workforce but reinforced the idea that they are essential in addressing care needs in areas with low concentrations of psychiatry, especially if they are authorized to work to the full extent of their training/education.
Creamer et al., 2017 [22]	Canadian Nurse Practitioner Core Competencies Identified: An Opportunity to Build Mental Health and Illness Skills and Knowledge	To guide future decisions on ANP entry-to-practice examinations and allow for Canadian Counsel of Registered Nurse Regulators member organizations to develop pan-Canadian requirements for licensure.	Descriptive, cross-sectional study with a mixed-methods approach	Primary care clinics and mental health clinics	Nurse practitioners	The study found inconsistent patterns of how psychiatric nurses are distributed relative to the rest of the workforce but reinforced the idea that they are essential in addressing care needs in areas with low concentrations of psychiatry, especially if they are authorized to work to the full extent of their training/education.
Curran, Mary-Jo et al., 2024 [23]	Facilitators of and Barriers to the Therapeutic Nurse-Patient Relationship: Perceptions From Psychiatric Mental Health Nurses	To explore the perspectives of psychiatric mental health nurses regarding factors that facilitate and impede the therapeutic nurse–patient relationship.	Cross-sectional design	Psychiatric and mental healthcare institutions	Registered nurses	The study highlighted various stakeholders’ calls for increased mental health education for nurse practitioners and identifies challenges and promising strategies for reaching this goal.
Delaney et al., 2018 [24]	The Effective Use of Psychiatric Mental Health Nurses in Integrated Care: Policy Implications for Increasing Quality and Access to Care	To implement integrated models of care where individuals’ medical and mental health needs are addressed holistically.	Mixed methods	Healthcare settings	Psychiatric mental health nurses, registered nurses, and advanced practice nurses	The study highlighted the importance of understanding facilitators and barriers in the therapeutic nurse–patient relationship.
Foster et al., 2019 [25]	Resilience and mental health nursing: An integrative review of international literature	To examine understandings and perspectives on resilience and explore and synthesize knowledge on resilience in mental health nursing.	Integrative review	International literature from a range of countries	Registered nurses and advanced psychiatric nurses	The study highlighted insufficient knowledge of the roles and skills of RNs and demonstrated how effective APNs can further the aims of integrated care models.
Hurley et al., 2022 [26]	Utilizing the mental health nursing workforce: A scoping review of mental health nursing clinical roles and identities	To collate and synthesize published research on the clinical roles of mental health nurses in order to systematically clarify their professional identity and potential.	Scoping review	International literature from a range of countries	Registered mental health nurses and registered psychiatric nurses	The study found that resilience has been variously constructed as an individual ability, a collective capacity, or as an interactive person–environment process.
Kilpatrick et al., 2024 [27]	A global perspective of advanced practice nursing research: A review of systematic reviews	To identify gaps in advanced practice nursing research globally.	Systematic review	International literature from multiple global regions	Individuals receiving care from advanced practice nurses, nurse practitioners, and clinical nurse specialists	The study revealed that the RNs perceive that APNs have a wide scope of technical skills, yet unmet needs among individuals receiving expert mental healthcare remain.
Phoenix et al., 2020 [28]	Effect of state regulatory environments on advanced psychiatric nursing practice	To examine how state and local regulation affects psychiatric mental health APN practice through the literature on how state scope-of-practice regulations affect the size and distribution of the broader APRN workforce.	Comparative cross-sectional study	Healthcare settings	Psychiatric nurse practitioners, clinical nurse specialists, registered nurses, and advanced psychiatric nurses	The study’s identified research gaps included interprofessional team functioning, workload, and patients and families as partners in healthcare.

## Data Availability

Data are contained within the article.

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
