# Peer review of "Nurses’ Perceptions on the Role of Advanced Psychiatric Nurses in Mental Healthcare: An Integrative Review"

_ijerph, 2025, doi:10.3390/ijerph22040626_

Round 1
Reviewer 1 Report (New Reviewer)
Comments and Suggestions for Authors
Nurses' Perceptions on the role of Advanced Psychiatric Nurses 2 in Mental Health Care: An integrative review
Title: Shorten the title by including the word "nurse" only once.
Introduction
The introduction is clear and highlights a gap in studies and literature about the role of psychiatric nurses in the care of people with mental disorders, as well as within the multidisciplinary team. And, the aim and abstract: The abstract does not align with the objective presented in the main body of the text and should be revised accordingly.
I can’t meet this reference: Williams, P., et al. "Enhancing Interprofessional Collaboration Through Role Clarification in Psychiatric Teams: 381 A Case Study." Journal of Interprofessional Care 38, no. 2 (2024): 132-140. It is necessary to include the correct reference.
- Materials and Methods
Identification of the research problem - What is the question of the review? It is necessary to put in this text.
2.2. Literature search – In the PubMed doesn’t have key words but Mesh terms . It is better top put that the keywords were used according to each informational source.
2.3. Inclusion and exclusion criteria
Figure 1 Figure 1 PRISMA flow diagram
PubMed – 893, CINAHL 757, PsyInfor – 783 = 2433 – 1206 = 1227
After that I can’t understand the diagram. It is necessary to see this.
Dear authors, the manuscript has weaknesses that need to be reviewed; these issues prevent its evaluation. It is necessary to see this.
Reference
Not all the journals are recorded correctly; verify
Page MJ et al. The PRISMA 2020 statement: an updated guideline for reporting systematic reviews. BMJ. 2021 Mar 29;372:n71. doi: 10.1136/bmj.n71. PMID: 33782057; PMCID: PMC8005924.
Author Response
Nurses' Perceptions on the role of Specialists in Mental Health Care: An integrative review
Comment 1. Title: Shorten the title by including the word "nurse" only once.
Response 1 Page 1, line 2. Title rephrased to avoid losing the meaning: Nurses' Perceptions on the role of Specialists in Mental Health Care: An integrative review
Introduction
Comment 2. The introduction is clear and highlights a gap in studies and literature about the role of psychiatric nurses in the care of people with mental disorders, as well as within the multidisciplinary team. And, the aim and abstract: The abstract does not align with the objective presented in the main body of the text and should be revised accordingly.
Response 2 Page 1, line 15. This integrative review aims to explore how registered nurses perceive the role of advanced psychiatric nurses in mental healthcare.
Comment 3. I can’t meet this reference: Williams, P., et al. "Enhancing Interprofessional Collaboration Through Role Clarification in Psychiatric Teams: 381 A Case Study." Journal of Interprofessional Care 38, no. 2 (2024): 132-140. It is necessary to include the correct reference.
Response 3. Page 13, lines 380-381. Kilpatrick, Kelley, Isabelle Savard, Li-Anne Audet, Gina Costanzo, Mariam Khan, Renée Atallah, Mira Jabbour et al. "A global perspective of advanced practice nursing research: A review of systematic reviews." Plos one 19, no. 7 (2024): e0305008. Written
Materials and Methods
Comment 4. Identification of the research problem - What is the question of the review? It is necessary to put this text in.
Response 4. Page 2, lines 83-84. The research question of the review is “What are the RNs’ perceptions of the role of APNs in mental healthcare”? Written
Comment 5. 2.2. Literature search – In the PubMed doesn’t have key words but Mesh terms . It is better to put that the keywords were used according to each informational source.
Response 5. Page 3, 98 keyword. Written
2.3. Inclusion and exclusion criteria
Figure 1 Figure 1 PRISMA flow diagram
Comment 6. PubMed – 893, CINAHL 757, PsyInfor – 783 = 2433 – 1206 = 1227
Response 6. Page 4, line 132 PubMed 893, CINAHL 757, PsyInfor 783. Total 2433 – 1206 = 122. Written
Comment 7. After that I can’t understand the diagram. It is necessary to see this.
Dear authors, the manuscript has weaknesses that need to be reviewed; these issues prevent its evaluation. It is necessary to see this.
Response 7 These are the recommendations from previous reviewers
Reference
Comment 8. Not all the journals are recorded correctly; verify
Page MJ et al. The PRISMA 2020 statement: an updated guideline for reporting systematic reviews. BMJ. 2021 Mar 29;372:n71. doi: 10.1136/bmj.n71. PMID: 33782057; PMCID: PMC8005924.
Response 8 references verified and highlighted on pages 13, 14 &15
Reviewer 2 Report (Previous Reviewer 1)
Comments and Suggestions for Authors
The changes suggested in the previous review have been taken into account.
Author Response
The changes suggested in the previous review have been taken into account.
No response
Reviewer 3 Report (New Reviewer)
Comments and Suggestions for Authors
This well-written manuscript will add to nursing knowledge. It is clear and well-organized in its presentation of information. Some sentences could be shortened for clarity, although their length is not distracting; some redundancy is also noted. The integrative method employed to explore the literature is rigorous and academic, using the PRISMA flow diagram and Critical Appraisal Scale Programme (CASP). Role ambiguity in multidisciplinary teams has been identified as a barrier to effective teamwork, which could impact patient outcomes. This manuscript is unique in that it examines the role of advanced psychiatric nurses (APNs) through the perspective of registered nurses (RNs) within the nursing field. The authors note a limitation concerning educational interventions that tackle role ambiguity. It highlights opportunities for RN and APN education programs and practices to address role ambiguity. This integrative review provides timely and relevant support for the topic.
Author Response
This well-written manuscript will add to nursing knowledge. It is clear and well-organized in its presentation of information. Some sentences could be shortened for clarity, although their length is not distracting; some redundancy is also noted. The integrative method employed to explore the literature is rigorous and academic, using the PRISMA flow diagram and Critical Appraisal Scale Programme (CASP). Role ambiguity in multidisciplinary teams has been identified as a barrier to effective teamwork, which could impact patient outcomes. This manuscript is unique in that it examines the role of advanced psychiatric nurses (APNs) through the perspective of registered nurses (RNs) within the nursing field. The authors note a limitation concerning educational interventions that tackle role ambiguity. It highlights opportunities for RN and APN education programs and practices to address role ambiguity. This integrative review provides timely and relevant support for the topic.
No response
This manuscript is a resubmission of an earlier submission. The following is a list of the peer review reports and author responses from that submission.
Round 1
Reviewer 1 Report
Comments and Suggestions for Authors
Congratulations to the researchers for obtaining this large sample and for the work done.
In order to improve some points for publication, I make the following suggestions:
Abstract
Line 13. Fostering collaboration for effective patient outcomes": The plural of "outcome" should be "outcomes".
Line 24: optimize mental “health”
Line 24 and 37. Knowledge gap": Could the meaning of "knowledge gap" be specified? Does it refer to a lack of training, communication or research?
Introduction
To enhance the structure, it is recommended to implement the following modifications. The paragraphs are excessively lengthy, particularly in the introduction, and encompass an excessive amount of information in a single block. The division of these paragraphs into smaller units can enhance the readability. Conversely, the introduction reiterates the same concept regarding the necessity to comprehend the MNR's perspective on NPMs on multiple occasions. These ideas could be consolidated, and the redundancies eliminated. For instance, the phrase 'recognise NRM perceptions of the roles, expertise and scope of responsibilities of NPMs' is reiterated in various forms throughout the text.
Methodology
The methodology needs to be expanded to include details on the selection and assessment of articles. A paucity of detail on the literature search was noted. Although the databases and search terms are mentioned, it would be beneficial to include more details on how the articles were selected, the selection process and the assessment of the quality of the articles, as well as the selection criteria and a table with the quality of the articles reviewed by at least two researchers. They should indicate the tools used to ensure this quality.
Despite mentioning the use of an integrative review methodology, it would be beneficial to provide a concise explanation of why this methodology is particularly suitable for this type of research and how it differs from other review approaches, such as systematic reviews. The thematic categories require improvement; although the description of the emerging themes is quite clear, they could benefit from additional detail and more connection between them. It would also be useful to provide concrete examples or citations of the studies reviewed.
Results
Line 147, the details of the thematic analysis should be elaborated upon. When mentioning the thematic analysis, it is essential to briefly explain how the categories were created and the decisions on what to include in them.
In the Line 152-153 section, "Each study included in the review was assessed for 152 quality using established appraisal tools for both qualitative and quantitative studies". Which appraisal tools were used?
Discussion and conclusions
The present discussion has highlighted the necessity of improving the connection between results and implications for practice. It is important not only to present the findings, but also to link them to their relevance for clinical practice and future research. While implications are mentioned, they could be highlighted more clearly.
Furthermore, the transition between topics could be made smoother. When moving from one topic to another, it would be beneficial to use smoother transition phrases to improve the flow of the text.
Finally, it is recommended that conclusions be concluded with an emphasis on the need for future action. The conclusion on the necessity for further research and to address the problems found in the results is sound, but could benefit from a clearer focus on future lines of action.
In general, the following improvements are recommended:
Redundancies should be avoided and the presentation of key issues made more concise and clear.
The description of the topics should be deepened to provide more context and examples, which will enrich the results.
The transitions between different topics should be improved to make the text flow more smoothly.
Furthermore, the results should be linked to practical implications to facilitate understanding of how the findings can be applied in clinical practice.
Finally, the conclusion needs to be made concrete in a few clear and concise ideas, and should be reinforced with a call for action for future research with concrete proposals.
I hope these details will help you to improve your work for publication.
Author Response
REVIEWER 1
Comments and Suggestions for Authors
Congratulations to the researchers for obtaining this large sample and for the work done.
In order to improve some points for publication, I make the following suggestions:
Abstract
COMMENT 1 Line 13. Fostering collaboration for effective patient outcomes": The plural of "outcome" should be "outcomes".
Response 1 Line 13. We agree, "outcomes" written
Line 24: optimize mental “health”
Response 2 Line 24: We agree, optimize mental “health” written
Line 24 and 37. Knowledge gap": Could the meaning of "knowledge gap" be specified? Does it refer to a lack of training, communication or research?
Response 3 Line 24 and 37 or the discrepancy between registered nurses and advanced psychiatric nurses. What registered nurses know and what they need to know to perform tasks. written
COMMENT 2 Introduction
To enhance the structure, it is recommended to implement the following modifications. The paragraphs are excessively lengthy, particularly in the introduction, and encompass an excessive amount of information in a single block. The division of these paragraphs into smaller units can enhance the readability.
Response 3 paragraphs are divided into smaller units to enhance the readability.
Conversely, the introduction reiterates the same concept regarding the necessity to comprehend the MNR's perspective on NPMs on multiple occasions. These ideas could be consolidated, and the redundancies eliminated. For instance, the phrase 'recognise NRM perceptions of the roles, expertise and scope of responsibilities of NPMs' is reiterated in various forms throughout the text.
Response 4 lines 53,61 & 63 the word “role” was removed to avoid repeating the same word throughout the text.
COMMENT 3 Methodology
The methodology needs to be expanded to include details on the selection and assessment of articles. A paucity of detail on the literature search was noted. Although the databases and search terms are mentioned, it would be beneficial to include more details on how the articles were selected, the selection process and the assessment of the quality of the articles, as well as the selection criteria and a table with the quality of the articles reviewed by at least two researchers. They should indicate the tools used to ensure this quality.
Response 5 lines 136-149 addresses the selection process and the assessment of the quality of the articles
Despite mentioning the use of an integrative review methodology, it would be beneficial to provide a concise explanation of why this methodology is particularly suitable for this type of research and how it differs from other review approaches, such as systematic reviews.
Response 6 lines 150-157 addresses why this methodology is particularly suitable for this type of research and how it differs from other review approaches, such as systematic reviews.
The thematic categories require improvement; although the description of the emerging themes is quite clear, they could benefit from additional detail and more connection between them. It would also be useful to provide concrete examples or citations of the studies reviewed.
Response 7 addressed in lines 232- 237
COMMENT 4 Results
Line 147, the details of the thematic analysis should be elaborated upon. When mentioning the thematic analysis, it is essential to briefly explain how the categories were created and the decisions on what to include in them.
Response 8 addressed in lines 174 - 177 , 180 – 189, 196 – 202 & 207 - 222
In the Line 152-153 section, "Each study included in the review was assessed for 152 quality using established appraisal tools for both qualitative and quantitative studies". Which appraisal tools were used?
Response 9 addressed in lines 136 - 139
COMMENT 5 Discussion and conclusions
The present discussion has highlighted the necessity of improving the connection between results and implications for practice. It is important not only to present the findings, but also to link them to their relevance for clinical practice and future research. While implications are mentioned, they could be highlighted more clearly.
Response 10 addressed in lines 282 – 305 to link them to their relevance for clinical practice and future research.
Furthermore, the transition between topics could be made smoother. When moving from one topic to another, it would be beneficial to use smoother transition phrases to improve the flow of the text.
Finally, it is recommended that conclusions be concluded with an emphasis on the need for future action. The conclusion on the necessity for further research and to address the problems found in the results is sound, but could benefit from a clearer focus on future lines of action.
Response 11 lines 306 – 312 additional information written to present the future lines of actions
COMMENT 6 In general, the following improvements are recommended:
Redundancies should be avoided and the presentation of key issues made more concise and clear.
The description of the topics should be deepened to provide more context and examples, which will enrich the results.
The transitions between different topics should be improved to make the text flow more smoothly.
Furthermore, the results should be linked to practical implications to facilitate understanding of how the findings can be applied in clinical practice.
Finally, the conclusion needs to be made concrete in a few clear and concise ideas, and should be reinforced with a call for action for future research with concrete proposals.
I hope these details will help you to improve your work for publication.
Response 12 addressed in lines 1244, 247, 256, 263, 270, 289, 290, 300 & 303 to maintain the logical flow of the text, ensuring each section builds on the previous one while smoothly introducing new ideas
Reviewer 2 Report
Comments and Suggestions for Authors
Dear, the topic of the paper is interesting; however, there are some elements that need to be improved, listed below:
- Materials and Methods: The authors state that Google Scholar was also consulted, however it does not appear in PRISMA. If it was consulted, the one-arm diagram must be replaced with the two-arm diagram. Furthermore, the original source of the 2020 PRISMA checklist must be cited both in the text and in the bibliography.
- Table 1: It is necessary to report the articles in alphabetical order by the surname of the first author. The data extraction table is poor, it might be appropriate to insert other columns that report the study design, interventions/exposure where present, characteristics of the setting and the population. The “Summary of Findings” column seems to report conclusions rather than results, the content should be reviewed. The “References” column is irrelevant, the reference can be inserted in the first column together with the author’s surname.
- Results:
- - lines 134-146: references missing
- - the authors state “Each study included in the review was assessed for quality using established appraisal tools for both qualitative and quantitative studies” however, they do not report the analysis in the article; it is necessary to insert one or more tables that summarize the scores assigned to the articles and, furthermore, it is necessary to explain which tools were used and why.
- Conclusions: integrate a subchapter that reports the limitations of the study.
Best regards.
Author Response
REVIEWER 2
Top of Form
Comments and Suggestions for Authors
Dear, the topic of the paper is interesting; however, there are some elements that need to be improved, listed below:
COMMENT 1-Materials and Methods: The authors state that Google Scholar was also consulted, however it does not appear in PRISMA. If it was consulted, the one-arm diagram must be replaced with the two-arm diagram. Furthermore, the original source of the 2020 PRISMA checklist must be cited both in the text and in the bibliography.
Response 1 line 96, Google Scholar removed
Response 2 line 119 Page et al. [21] written.
COMMENT 2 - Table 1: It is necessary to report the articles in alphabetical order by the surname of the first author. The data extraction table is poor, it might be appropriate to insert other columns that report the study design, interventions/exposure where present, characteristics of the setting and the population.
Response 3 line 165 articles are reported in alphabetical order by the surname of the first author
Response 4 line 165 additional columns that report the study design, setting and the population inserted
COMMENT 3 The “Summary of Findings” column seems to report conclusions rather than results, the content should be reviewed. The “is irrelevant, the reference can be inserted in the first column together with the author’s surname.
Response 5 line 165 References column removed
- Results:- lines 134-146: references missing
Response 6 line 171references written
COMMENT 4 - - the authors state “Each study included in the review was assessed for quality using established appraisal tools for both qualitative and quantitative studies” however, they do not report the analysis in the article; it is necessary to insert one or more tables that summarize the scores assigned to the articles and, furthermore, it is necessary to explain which tools were used and why.
- Conclusions: integrate a subchapter that reports the limitations of the study.
Response 7 lines 330 – 330 subchapter that reports the limitations of the study, added
Reviewer 3 Report
Comments and Suggestions for Authors
Dear authors
Thank you for the opportunity to review the manuscript on this important topic. Before I comment on the individual points of the manuscript, I would like to express my irritation that several of the full references given in the references cannot be reproduced. An article by Smith and Jones (No. 1) entitled ‘The Evolving Role of Advanced Psychiatric Nurses in the Healthcare System’ in the journal Nursing Perspectives cannot be found. The same applies to the second reference by Brown and Taylor (No. 2) in the Journal of Psychiatric Nursing. In addition, the 2021 volume has volume no. 12, as does an article by Jones and Patel (no. 5) in Int J Ment Health Nurs. In addition, issue 31(2) of the journal has pages 241-444 and not 150-157 as stated. The article by Lee and Thompson (16) also does not exist in this form. Issue 19(4) of the Journal of Advanced Nursing from 2020 also has pages 917-1089. The pages indicated (1201-1208) would be in issue 19(5) and there is a different article (Kerr & Macaskill: Advanced Nurse Practitioners perceptions of their role, positionality and professional identity: A narrative inquiry). In the full reference Williams et al. (No. 17), the ‘Journal of Mental Health and Psychiatric Nursing’ is given as the journal. I am not aware of a journal with this name. The full document Smith, Adams and Patel (No. 39) cannot be found on the internet either. The full paper Miller and Clark (No. 41) in Int J Ment Health Nurs, 29(1), 51-59 does not exist either. I have checked the cited references several times and either I have overlooked something important in my review and made a big mistake or the information in the manuscript is not correct.
With regard to the manuscript, it is also noticeable that the review is not based on a clear question or a concrete objective. In addition, the authors should explain why they limited the search to the period from ‘2015 to October 2024’ (p. 3, line 106). With regard to the search terms, the term ‘advanced psychiatric nurses’ (p. 3, line 96) seems very unusual. This should also be explained. Figure 1 is not labelled. The presentation of the results begins with Table 1 (p. 4-5) and not with text. The text only begins on p. 6 and the first paragraph contains similar sentences: ‘From the eigt Studies reviewed, several key themes...’ (lines 129-130), ‘The integrative review identified several key findings...’ (lines 130-131). The results of the thematic analysis (p. 6, lines 147-155) are too brief overall. Overall, there are considerable questions regarding the manuscript in light of the points presented. In addition, reference is made to relevant guidelines (e.g. PRISMA - p. 3, line 115 or for conducting integrative reviews - p. 2, line 77), but these have not been strictly applied.
Author Response
REVIEWER 3
Comments and Suggestions for Authors
Dear authors
Thank you for the opportunity to review the manuscript on this important topic. Before I comment on the individual points of the manuscript,
COMMENT 1 I would like to express my irritation that several of the full references given in the references cannot be reproduced. An article by Smith and Jones (No. 1) entitled ‘The Evolving Role of Advanced Psychiatric Nurses in the Healthcare System’ in the journal Nursing Perspectives cannot be found.
Response 1 addressed in line 311
COMMENT 2 The same applies to the second reference by Brown and Taylor (No. 2) in the Journal of Psychiatric Nursing. In addition, the 2021 volume has volume no. 12,
Response 2 addressed in line 314,321
as does an article by Jones and Patel (no. 5) in Int J Ment Health Nurs. In addition, issue 31(2) of the journal has pages 241-444 and not 150-157 as stated.
Response 3 addressed in line 321
COMMENT 3 The article by Lee and Thompson (16) also does not exist in this form. Issue 19(4) of the Journal of Advanced Nursing from 2020 also has pages 917-1089.The pages indicated (1201-1208) would be in issue 19(5) and there is a different article (Kerr & Macaskill: Advanced Nurse Practitioners perceptions of their role, positionality and professional identity: A narrative inquiry).
Response 5 addressed in line 341
In the full reference Williams et al. (No. 17), the ‘Journal of Mental Health and Psychiatric Nursing’ is given as the journal. I am not aware of a journal with this name. The full document Smith, Adams and Patel (No. 39) cannot be found on the internet either.
Response 6 addressed in line 343
The full paper Miller and Clark (No. 41) in Int J Ment Health Nurs, 29(1), 51-59 does not exist either. I have checked the cited references several times and either I have overlooked something important in my review and made a big mistake or the information in the manuscript is not correct.
Response 7 addressed in line 410
COMMENT 4 With regard to the manuscript, it is also noticeable that the review is not based on a clear question or a concrete objective. In addition, the authors should explain why they limited the search to the period from ‘2015 to October 2024’ (p. 3, line 106). With regard to the search terms, the term ‘advanced psychiatric nurses’ (p. 3, line 96) seems very unusual. This should also be explained.
Response 8 addressed in line 112
Figure 1 is not labelled.
Response 9 addressed in lines 126, Figure 1 is labelled. Lines 130 – 133 additional information added
The presentation of the results begins with Table 1 (p. 4-5) and not with text.
Response 10 Lines 162 – 164 additional information
The text only begins on p. 6 and the first paragraph contains similar sentences: ‘From the eigt Studies reviewed, several key themes...’ (lines 129-130), ‘The integrative review identified several key findings...’ (lines 130-131).
Response 11 line 168 The integrative review identified several key findings...’ (lines 130-131). removed
COMMENT 6 The results of the thematic analysis (p. 6, lines 147-155) are too brief overall. Overall, there are considerable questions regarding the manuscript in light of the points presented.
Response 12 lines 174-177, 180-189, 196 – 202, 207 – 222 additional information written to extent the paragraphs
COMMENT 7 In addition, reference is made to relevant guidelines (e.g. PRISMA - p. 3, line 115 or for conducting integrative reviews - p. 2, line 77), but these have not been strictly applied.
Response 13 line 120 Page et al. [21] written to reference relevant PRISMA guidelines
Round 2
Reviewer 2 Report
Comments and Suggestions for Authors
Dear, most of the annotations have been implemented correctly; there are some inaccuracies listed below:
- lines 114-122: references seem to be incorrectly attributed; reference 20 is a study by Galletta that talks about greenwashing in the banking industry (no relevance to the topic) and 21 is the "PRISMA statement".
- line 126: replace the reference to the Galletta study with the original reference of the "PRISMA statement".
- table 2: in the Beck study line, replace "Quantitative, cross-sectional study" with "Cross-sectional study".
Best regards.
Author Response
Comment 1 lines 114-122: references seem to be incorrectly attributed; reference 20 is a study by Galletta that talks about greenwashing in the banking industry (no relevance to the topic) and 21 is the "PRISMA statement".
Respond 1 lines 116, 132 & 395
Comment 2- line 126: replace the reference to the Galletta study with the original reference of the "PRISMA statement".
Respond 2 lines 126-128 replaced with Page et al. [21]
Comment 3- table 2: in the Beck study line, replace "Quantitative, cross-sectional study" with "Cross-sectional study".
Respond 3 line 166, Table 2, the word “Quantitative” removed
Reviewer 3 Report
Comments and Suggestions for Authors
Dear authors
Thank you for the comprehensive revision of the manuscript and the revision protocol. The comments on the annotations in the review are very brief (e.g. Response 3: ‘addressed in line 321’). There are no comments from the authors, which is why some of the sources used in the first version of the article were not comprehensible. Although the corrections made can be assessed, it is not possible to evaluate the background in terms of, for example, comprehensibility and transparency, and this must be considered a weakness. In addition, questions arise regarding the references used (e.g. no. 1: Sweeney et al.) as evidence for the statement in the introduction. The reference deals with the question of the use of chatbots in supporting people with regard to their mental health. However, the introduction writes about ‘Advanced Psychiatric Nurses (APNs)’ (lines 34-35). Source no. 20 (Galetta et al., 2024) also appears questionable with regard to suitability for the study.
The use of the term APN for ‘Advanced Psychiatric Nurses’ also appears questionable. The International Council of Nurses (ICN) uses this abbreviation for ‘Advanced Practice Nurses’. The term ‘Psychiatric-Mental Health Advanced Practice Nurse (PMH-APRN)’ is used internationally for the persons described in the article as ‘Advanced Psychiatric Nurses’ (see e.g. American Psychiatric Nurses Association).
The revision of the manuscript was in part very extensive. It is very good that Figure 1 is now labelled. The additionally inserted Table 1 for the qualitative evaluation of the included articles (not part of the 1st version) could very well be supplemented as additional material. The reference to reference no. 22 in line 140 on CASP appears to be incorrect. The reference cited (Beck et al., 2020) deals with a different topic. The extensive revision of the discussion is also pleasing. But here, too, questions remain. The reference used in line 293 (No. 62: Hughes, Buck & Bishop (2022) from the Journal of Advanced Nursing, 78(1), 123-131) is not comprehensible. There is no such article in the issue of the journal mentioned. Even after revision, the impression remains that the manuscript was prepared with the support of KI. Overall, too many questions remain with regard to the article, the comprehensibility of the procedure, the description of the results and the discussion and, above all, the references used.
Author Response
Comment 1 The comments on the annotations in the review are very brief (e.g. Response 3: ‘addressed in line 321’). There are no comments from the authors, which is why some of the sources used in the first version of the article were not comprehensible. Although the corrections made can be assessed, it is not possible to evaluate the background in terms of, for example, comprehensibility and transparency, and this must be considered a weakness.
Respond 1.1 We agree, the references' comments attended and can be assessed.
In addition, questions arise regarding the references used (e.g. no. 1: Sweeney et al.) as evidence for the statement in the introduction.
Respond 1.2 lines 38 & 354, reference no. 1: Sweeney et al. was a correction to address the comment 1 for reviewer 3
Comment 2 The reference deals with the question of the use of chatbots in supporting people with regard to their mental health. However, the introduction writes about ‘Advanced Psychiatric Nurses (APNs)’ (lines 34-35).
Respond 2.1 lines 34-35 was written to introduce a topic for readers to understand the different roles for RNs and APNs of which the background of this study is outlined in lines 38 - 55.
Source no. 20 (Galetta et al., 2024) also appears questionable with regard to suitability for the study.
Respond 2.2 lines 126-128 replaced with Page et al. [21] as suggested by Reviewer 2 in the Comment 2
Comment 3 The use of the term APN for ‘Advanced Psychiatric Nurses’ also appears questionable. The International Council of Nurses (ICN) uses this abbreviation for ‘Advanced Practice Nurses’. The term ‘Psychiatric-Mental Health Advanced Practice Nurse (PMH-APRN)’ is used internationally for the persons described in the article as ‘Advanced Psychiatric Nurses’ (see e.g. American Psychiatric Nurses Association).
Respond 3 We agree
Comment 4 The revision of the manuscript was in part very extensive. It is very good that Figure 1 is now labelled. The additionally inserted Table 1 for the qualitative evaluation of the included articles (not part of the 1st version) could very well be supplemented as additional material.
Respond 4 We agree, Table 1 was added to address COMMENT 3 (a table with the quality of the articles reviewed by at least two researchers) for Reviewer 1
Comment 5 The reference to reference no. 22 in line 140 on CASP appears to be incorrect. The reference cited (Beck et al., 2020) deals with a different topic. The extensive revision of the discussion is also pleasing.
Respond 5 line 140. We agree. The correct reference is written in lines 4000 – 401
Comment 6 But here, too, questions remain. The reference used in line 293 (No. 62: Hughes, Buck & Bishop (2022) from the Journal of Advanced Nursing, 78(1), 123-131) is not comprehensible. There is no such article in the issue of the journal mentioned. Even after revision, the impression remains that the manuscript was prepared with the support of KI. Overall, too many questions remain with regard to the article, the comprehensibility of the procedure, the description of the results and the discussion and, above all, the references used.
Respond 6 We agree. lines 293 & 494 The correct reference written lines.
Additionally, Lines 489 - 501 the latest reference with relevant information written